# BRCA Mutation Status in Triple-Negative Breast Cancer Patients Treated with Neoadjuvant Chemotherapy: A Pivotal Role for Treatment Decision-Making

**DOI:** 10.3390/cancers14194571

**Published:** 2022-09-21

**Authors:** Francesco Pavese, Ettore Domenico Capoluongo, Margherita Muratore, Angelo Minucci, Concetta Santonocito, Paola Fuso, Paola Concolino, Enrico Di Stasio, Luisa Carbognin, Giordana Tiberi, Giorgia Garganese, Giacomo Corrado, Alba Di Leone, Daniele Generali, Simona Maria Fragomeni, Tatiana D’Angelo, Gianluca Franceschini, Riccardo Masetti, Alessandra Fabi, Antonino Mulè, Angela Santoro, Paolo Belli, Giampaolo Tortora, Giovanni Scambia, Ida Paris

**Affiliations:** 1Division of Oncological Gynecology, Department of Women’s and Children’s Health, Fondazione Policlinico Universitario A. Gemelli IRCCS, 00168 Rome, Italy; 2Department of Molecular Medicine and Medical Biotechnology, Federico II University, 80131 Naples, Italy; 3Department of Clinical Pathology and Genomics, Cannizzaro Hospital, 95126 Catania, Italy; 4Clinical Chemistry, Biochemistry and Molecular Biology Operations (UOC), Fondazione Policlinico Universitario A. Gemelli IRCCS, 00168 Rome, Italy; 5Gynaecology and Breast Care Center, Mater Olbia Hospital, 07026 Olbia, Italy; 6Dipartimento Universitario Scienze della Vita e Sanità Pubblica, Sezione di Ginecologia ed Ostetricia, Università Cattolica del Sacro Cuore, 00168 Rome, Italy; 7Department of Medical, Surgery and Health Sciences, University of Trieste, 34149 Trieste, Italy; 8Unit of Precision Medicine in Breast Cancer, Scientific Directorate, Department of Woman and Child Health and Public Health, Fondazione Policlinico Universitario A. Gemelli IRCCS, 00168 Rome, Italy; 9Unità di Gineco-Patologia e Patologia Mammaria, Dipartimento Scienze della Salute della Donna, Fondazione Policlinico Universitario A. Gemelli IRCCS, 00168 Rome, Italy; 10UOC Radiologia Generale ed Interventistica Generale, Area Diagnostica per Immagini, Dipartimento Diagnostica per Immagini, Radioterapia Oncologica ed Ematologia, Fondazione Policlinico Universitario A. Gemelli IRCCS, 00168 Rome, Italy; 11Comprehensive Cancer Center, Unit of Medical Oncology, Fondazione Policlinico Universitario A. Gemelli IRCCS, 00168 Rome, Italy; 12Medical Oncology, Università Cattolica del Sacro Cuore, 00168 Rome, Italy

**Keywords:** triple-negative breast cancer, BRCA1/2, neoadjuvant chemotherapy, platinum agents

## Abstract

**Simple Summary:**

In this retrospective observational study, we evaluated data from patients with triple-negative breast cancer (TNBC) treated with neoadjuvant chemotherapy (NACT) in order to better define the impact of germline BRCA1/2 (gBRCA1/2) mutation status on outcomes in this patient population. Our results show that patients with BRCA1/2 mutation had a higher pathologic complete response (pCR) rate than non-mutated patients; nevertheless, the benefit was confirmed only in the subset of patients who received a platinum-based NACT. Furthermore, pCR was associated with improved Event Free Survival (EFS) and Overall Survival (OS), regardless of BRCA1/2 mutation status and type of NACT received. Long-term follow-up analyses are needed to further define the impact of gBRCA mutation status in patients with early-TNBC.

**Abstract:**

Triple-negative breast cancer (TNBC) is characterized by earlier recurrence and shorter survival compared with other types of breast cancer. Moreover, approximately 15 to 25% of all TNBC patients harbor germline BRCA (gBRCA) 1/2 mutations, which confer a more aggressive phenotype. However, TNBC seems to be particularly sensitive to chemotherapy, the so-called ‘triple negative paradox’. Therefore, Neoadjuvant chemotherapy (NACT) is currently considered the preferred approach for early-stage TNBC. BRCA status has also been studied as a predictive biomarker of response to platinum compounds. Although several randomized trials investigated the addition of carboplatin to standard NACT in early-stage TNBC, the role of BRCA status remains unclear. In this retrospective analysis, we evaluated data from 136 consecutive patients with Stage I-III TNBC who received standard NACT with or without the addition of carboplatin, in order to define clinical features and outcomes in BRCA 1/2 mutation carriers and non-carrier controls. Between January 2013 and February 2021, 67 (51.3%) out of 136 patients received a standard anthracyclines/taxane regimen and 69 (50.7%) patients received a platinum-containing chemotherapy regimen. Deleterious germline BRCA1 or BRCA2 mutations were identified in 39 (28.7%) patients. Overall, patients with deleterious gBRCA1/2 mutation have significantly higher pCR rate than non-carrier patients (23 [59%] of 39 vs. 33 [34%] of 97; *p* = 0.008). The benefit of harboring a gBRCA mutation was confirmed only in the subset of patients who received a platinum-based NACT (17 [65.4%] of 26 vs. 13 [30.2%] of 43; *p* = 0.005) while no differences were found in the platinum-free subgroup. Patients who achieved pCR after NACT had significantly better EFS (OR 4.5; 95% CI 1.9–10.7; *p* = 0.001) and OS (OR 3.3; 95% CI 1.3–8.9; *p* = 0.01) than patients who did not, regardless of BRCA1/2 mutation status and type of NACT received. Our results based on real-world evidence show that TNBC patients with the gBRCA1/2 mutation who received platinum-based NACT have a higher pCR rate than non-carrier patients, supporting the use of this chemotherapy regimen in this patient population. Long-term follow-up analyses are needed to further define the role of gBRCA mutation status on clinical outcomes in patients with early-TNBC.

## 1. Introduction

Triple-negative breast cancer (TNBC) accounts for 10–20% of invasive breast cancers, carrying a poorer prognosis than other BCs subtypes [1,2,3].

The pathological definition is characterized by the absence of estrogen receptor (ER) and progesterone receptor (PR) expression, and human epidermal growth factor receptor 2 (HER2) expression or amplification. Nevertheless, TNBC is a highly heterogeneous disease: Based on gene expression profiles, six TNBC subtypes were identified (Basal-like 1 and 2, Immunomodulatory, Mesenchymal, Mesenchymal stem-like, Luminal androgen receptor) displaying unique biology and drug sensitivities [4]. Moreover, approximately 15 to 25% of all TNBC patients harbor germline BRCA (gBRCA) 1/2 mutations [5,6,7] conferring a more aggressive phenotype. This subset is associated with Hereditary Breast/Ovarian cancer Syndrome, predisposing women to both breast and ovarian cancer [8,9].

Despite the aggressive clinical behavior, a subset of TNBCs seems to be particularly sensitive to chemotherapy in both advanced and early stages, the so-called ‘triple negative paradox’ [10,11].

The neoadjuvant approach in breast cancer is a perfect model to test the chemosensitivity of the tumor in vivo and adequately guides the type of treatment by monitoring the response in time-course [12]. Patients achieving pCR after neoadjuvant treatment have been shown to have improved event-free survival (EFS) and overall survival (OS) compared with those with residual tumors at the time of surgery. The association between pCR and long-term outcomes was strongest in patients with aggressive tumor subtypes such as TNBCs [13,14]. Therefore, neoadjuvant chemotherapy (NACT) with anthracyclines and taxanes is currently considered the preferred approach for the majority of TNBC patients with early-stage disease [15].

Several preclinical studies have suggested that BRCA1-related BC cell lines show a varying degree of sensitivity to different chemotherapeutic agents [16,17,18,19], explaining the distinct pathologic features in hereditary BC compared with sporadic cancer [20,21]. Particularly, DNA-repair defects, which are characteristic of BRCA-related cancers, were shown to confer sensitivity to DNA-damaging agents, such as platinum derivatives and poly (ADP ribose) polymerase (PARP) inhibitors [16,22,23].

Regardless of this strong biological rationale, the role of platinum compounds in BRCA-mutated patients remains controversial, and unfortunately, the most effective chemotherapy regimen in gBRCA mutation carriers is still under debate [24,25,26].

Recently, several RCTs have investigated the possible benefit of adding a platinum agent to standard NACT in TNBC patients [27,28,29]. Despite platinum-based NACT having been associated with significantly increased pCR rates, results on survival outcomes were controversial [30,31]. Moreover, data on gBRCA mutation carriers are inconclusive due to the small number of patients included in the studies [32].

Our single-institution study reports data collected retrospectively in TNBC patients treated with NACT to assess the impact of germline BRCA1/2 mutation status on clinical outcomes in this patient population.

## 2. Materials and Methods

### 2.1. Study Design and Patient Recruitment

From January 2013 to February 2021, we evaluated data from 136 consecutive women with stage I-III TNBC who received NACT at the Gynecologic Oncology Division, Fondazione Policlinico A. Gemelli IRCCS, Rome (Italy).

All patients had measurable breast disease documented by a clinical evaluation, mammogram, ultrasound, and magnetic resonance imaging (MRI) of breasts at basal and after treatment.

Samples of diagnostic core-biopsy were obtained from each patient and examined by designated breast pathologists. Triple-negative status was defined as less than 1% positivity for estrogen receptor and progesterone receptor expression by immunohistochemical staining, and either a score of 0–1+ in HER2 staining by immunohistochemistry or no HER2 gene amplification by fluorescent in situ hybridization [33].

All patients underwent genetic testing for BRCA 1 and BRCA 2 germline mutations, employing next-generation sequencing (NGS) techniques. All variants identified in both genes were analyzed, carefully examined, and classified according to several databases including ClinVar, LovD, and UMD and in reference to the literature.

Patients receiving NACT have been treated with two regimen approaches before curative surgery: (a) Anthracycline (epirubicin 90 mg/m^2^ or adriamycin 60 mg/m^2^) plus cyclophosphamide (600 mg/m^2^) intravenously every 2–3 weeks for four cycles, followed by paclitaxel (80 mg/m^2^) intravenously once a week for twelve weeks; (b) carboplatin (Area Under Curve (AUC) 1.5) plus paclitaxel (80 mg/m^2^) intravenously once a week for twelve weeks, followed by anthracycline (epirubicin 90 mg/m^2^ or adriamycin 60 mg/m^2^) plus cyclophosphamide (600 mg/m^2^) intravenously every 2–3 weeks for four cycles. Adjuvant radiotherapy was performed according to international guidelines; it is worth noting that, recently, in the absence of pCR after NACT, adjuvant treatment with capecitabine for six to eight cycles has been introduced as a standard of practice [25].

The primary outcome was to evaluate the pCR rate in TNBC patients receiving NACT according to gBRCA mutational status. The secondary endpoints included Event Free Survival (EFS) and Overall Survival (OS). The analysis was performed according to disease characteristics and the type of administered NACT.

pCR was defined as the absence of residual invasive disease with or without ductal carcinoma in situ and the absence of any tumor deposit ≥0.2 mm in sampled axillary nodes (ypT0/is ypN0).

EFS was defined as the interval from diagnostic core-biopsy to the earliest occurrence of disease progression resulting in inoperability, locoregional recurrence (after neoadjuvant therapy), distant metastases, or death from any cause. Patients alive without an event as of the analysis cutoff date were censored at the last study follow-up date. OS was defined as the interval from diagnostic core-biopsy to death. For patients alive on the data cutoff date, survival was censored at the last study follow-up date.

The Ethics Committee of Fondazione Policlinico A. Gemelli IRCCS of Rome (Italy) approved the clinical trial, and written informed consent was obtained from all patients.

### 2.2. Statistical Analysis

Analyses were performed using IBM SPSS Statistics software, Version 24.0 (IBM Corp. Released 2016. IBM SPSS Statistics for Windows, Version 24.0. Armonk, NY, USA: IBM Corp.).

All data were first analyzed for normality of distribution using the Kolmogorov–Smirnov test of normality. Continuous variables were expressed as mean ± SD, unless otherwise specified, categorical variables were displayed as frequencies, and the appropriate parametric (*t*-test) or non-parametric test (Mann–Whitney or χ^2^-test) was used to assess the significance of the differences between subgroups. Kaplan–Meier methods were employed to generate survival plots that were compared using a log-rank test. Multivariate Cox proportional hazards regression was employed to evaluate survival models with calculated odds ratios (ORs) and 95% CIs. Two-sided *p*-values were considered to be statistically significant if <0.05.

## 3. Results

Between January 2013 and February 2021, we evaluated 136 patients with primary invasive TNBC, who received NACT at the Gynecologic Oncology Division, Fondazione Policlinico A. Gemelli IRCCS, Rome (Italy).

The demographic and disease characteristics of the patients are summarized in Table 1.

In total, 67 (51.3%) out of 136 patients received the standard anthracyclines/taxane regimen; 69 (50.7%) patients received a platinum-containing chemotherapy regimen. Deleterious germline BRCA1 or BRCA2 mutations were identified in 39 out of 136 (28.7%) patients. Moreover, the variants of uncertain significance (VUS) were found in nine patients, and thus were included in the cohort of patients with BRCA1/2 wild-type. BRCA1/2-related BCs had similar pathological characteristics with regards to the sporadic tumors in terms of histological grade, lymph node involvement, and Ki-67 level, as described in Table 2. The gBRCA1/2 mutations and their related disease characteristics are listed in Table 3. The classification of BRCA gene variants is described according to the variant classification guidelines published by Richards et al. in 2015 [34]. This classification is applicable to variants in all Mendelian genes and comprises a five-tier system of classification for variants relevant to Mendelian disease.

A platinum-containing chemotherapy regimen was administered in 26 (66.7%) patients with a deleterious germline BRCA1/2 mutation and in 43 (44.3%) patients without a deleterious germline BRCA1/2 mutation. Median follow-up was 38 months [interquartile range, 22–55 months].

In total, 56 (41.2%) out of 136 enrolled patients achieved pCR. The proportion of patients achieving pCR was significantly higher among BRCA1/2 mutation carriers than among non-carriers (23 [59%] of 39 vs. 33 [34%] of 97; OR 2.78, 95% CI 1.29–5.98; *p* = 0.0085). Among patients who received platinum-containing NACT, 17 (65.4%) out of 26 patients with a deleterious germline BRCA1/2 mutation achieved pCR vs. 13 (30.2%) out of 43 patients without a germline BRCA1/2 mutation (OR 4.35, 95% CI 1.54–12.30; *p* = 0.0054). In 67 patients treated with A + T NACT, 6 (46.2%) out of the 13 were carrying the BRCA1/2 mutation; on the other hand, 20 (37.0%) out of the 54 with gBRCA wt achieved pCR (*p* = 0.545) (Figure 1).

Overall, no significant difference in the pCR rate was observed between patients receiving a platinum-based NACT and those receiving a platinum-free regimen (30 [43.5%] of 69 vs. 26 [38.8%] of 67; OR 1.21; 95% CI 0.61–2.4; *p* = 0.58).

Among patients with a deleterious germline BRCA1/2 mutation, 17 (65.4%) out of 26 patients who received carboplatin achieved pCR compared with 6 (46.2%) of 13 patients who did not (OR 2.2; 95% CI 0.56–8.56; *p* = 0.25). Of the 96 patients without a deleterious germline BRCA1/2 mutation, 13 (30.2%) of 43 patients who received carboplatin and 20 (37.0%) of 54 patients who received platinum-free NACT achieved pCR (OR 0.73; 95% CI 0.31–1.72; *p* = 0.48).

Patients with stage I tumors showed a higher pCR rate than those with stage III tumors (OR 6.00; 95% CI 1.69–21.21; *p* = 0.0054), while no differences were found compared to patients with stage II cancers. No other significant differences were observed with regard to lymph node status and histological grade (Table 4).

The estimated mean EFS for gBRCA1/2 mutation carriers was 73.7 ± 4.9 months compared to 59.3 ± 3.5 months for non-carriers (*p* = 0.123).

Among patients with the gBRCA1/2 mutation, the estimated mean OS was 101.1 ± 6.3 months vs. 66.5 ± 2.9 months among non-mutated patients (*p* = 0.127).

Overall, the estimated mean EFS was 59.6 ± 2.7 months among patients treated with platinum-based NACT, compared with 61.1 ± 4.2 months among patients treated with a platinum-free regimen (*p* = 0.079).

In 39 patients harboring a gBRCA1/2 mutation, two (7%) events were observed in patients who received a platinum-based regimen, whereas five (38%) events were observed in the platinum-free group.

Patients who achieved pCR had improved EFS and OS compared with those who did not. The estimated mean EFS was 70.8 ± 2.8 months among patients who attained pCR, compared with 57.1 ± 4.3 months among patients who did not (*p* = 0.0002). The estimated mean OS was 103.7 ± 4.4 months among patients who attained pCR, compared with 67.9 ± 3.8 months among patients who did not (*p* = 0.01).

Using the multivariate Cox regression analysis, pCR was confirmed as the only variable associated with improved EFS (OR 4.5; 95% CI 1.9–10.7; *p* = 0.001) and OS (OR 3.3; 95% CI 1.3–8.9; *p* = 0.016) (Figure 2).

## 4. Discussion

In this retrospective analysis, we investigated whether the presence of a deleterious gBRCA1/2 mutation could affect the response to treatment and impact clinical outcomes such as EFS and OS in TNBC patients receiving NACT.

In our study population, the percentage of patients harboring a gBRCA1/2 mutation was slightly higher than expected (28.6%). However, our department is also a referral center for the treatment of ovarian cancer, therefore a large number of BRCA1/2 mutation carriers (affected and non-affected) perform the breast screening program at our institute.

Overall, our study showed that patients with a deleterious gBRCA1/2 mutation have significantly higher pCR rates than BRCA1/2 wild-type patients. The clinical benefit of harboring a gBRCA mutation was confirmed in the subset of patients who received platinum-based NACT, but not among those who received a platinum-free regimen. These findings are consistent with the aforementioned observation that BRCA1/2 mutation-associated cancers are particularly sensitive to DNA crosslinking agents such as platinum salts [16].

To date, however, available clinical data in this setting are limited and controversial. In the neoadjuvant setting, several trials showed platinum-based chemotherapy is an active regimen in BRCA-mutated BC patients, especially in those affected by TNBC [36,37,38].

In line with our findings, Wunderle et al. (2018) reported the results of a retrospective analysis of 355 BC patients treated with NACT with or without the addition of carboplatin. Overall, the pCR rate was greater in gBRCA mutation carriers irrespective of tumor subtype (54.3% vs. 22.6%). In addition, the highest pCR rate was observed in TNBC with the gBRCA1/2 mutation who received carboplatin (73.3%) [36].

Accordingly, Wang et al. (2015) observed, in a large retrospective cohort study, that among TNBC patients receiving NACT with or without the addition of carboplatin, gBRCA1 mutation carriers exhibited a higher pCR rate than non-carriers (53.8% vs. 29.7%). Even after adjusting for other characteristics (lymph node status and tumor grade), BRCA1 mutation status remained an independent significant predictor of pCR [37].

Similar findings were reported by Sella et al. (2018) who observed that, in a population of 119 TNBC patients receiving a platinum-based or a platinum-free NACT, gBRCA mutation carriers achieved a higher pCR rate compared with non-carriers (64.3% vs. 44.8%). On multivariate analysis, while no benefit was observed from the addition of carboplatin, BRCA mutation status was significantly associated with pCR (HR 4.00, 95%CI 1.65–9.75, *p* = 0.002) [38].

However, only two RCTs that investigated the efficacy of platinum-based vs. platinum-free NACT in TNBC patients had reported pCR results according to gBRCA mutational status [29,39]. The GeparSixto GBG66 trial evaluated the additional effect of carboplatin in NACT containing an anthracycline, a taxane, and a targeted therapy (trastuzumab/lapatinib or bevacizumab) on pCR in patients with stage II–III TNBC and HER2-positive BC. In total, 315 patients with TNBC received a non-traditional regimen of weekly paclitaxel (80 mg/m^2^), non-pegylated liposomal doxorubicin (20 mg/m^2^) once per week, and bevacizumab (15 mg/kg) every 3 weeks, with or without weekly carboplatin (AUC 1.5–2) for 18 weeks. In this subset, the addition of neoadjuvant carboplatin significantly increased the pCR rate from 42.7% to 57%. The secondary analysis of this study showed that TNBC patients with the gBRCA1/2 mutation achieved a superior pCR rate compared to non-mutated patients, although no benefit from the addition of carboplatin was observed [39]. Similarly, the BrighTNess trial assessed the addition of the PARP inhibitor veliparib plus carboplatin or carboplatin alone compared to standard NACT in stage II-III TNBC patients. In total, 634 patients were randomized to receive one of three regimens: Weekly paclitaxel (80 mg/m^2^) for 12 doses plus carboplatin (AUC 6) every 3 weeks for four cycles plus veliparib (50 mg orally twice a day); paclitaxel plus carboplatin plus veliparib placebo (twice a day); paclitaxel plus carboplatin placebo (every 3 weeks for four cycles) plus veliparib placebo. All patients subsequently received doxorubicin (60 mg/m^2^) and cyclophosphamide (600 mg/m^2^) every 2–3 weeks for four cycles. Results showed that the proportion of patients achieving pCR was significantly improved in the veliparib plus carboplatin plus paclitaxel group compared with the paclitaxel-based chemotherapy-alone group (53% vs. 31%, *p* < 0.0001), whereas the addition of veliparib to neoadjuvant carboplatin plus paclitaxel did not seem to increase the proportion of patients who achieved pCR compared with carboplatin plus paclitaxel alone (53% vs. 58%, *p* = 0.36). In this RCT, the addition of carboplatin increased the pCR rate in the subset of patients with a gBRCA mutation (57% in the paclitaxel plus carboplatin plus veliparib group vs. 50% in the paclitaxel plus carboplatin group vs. 41% in the paclitaxel alone group) but the study was not powered to show significant differences in the attainment of pCR with the addition of carboplatin, with or without veliparib [29].

Pooled results from both RCTs showed that the addition of carboplatin to anthracycline and taxane-based NACT was not associated with a significantly higher rate of pCR in BRCA-mutated BC patients (OR 1.17, 95%CI 0.51–2.67, *p* = 0.711) while the benefit was present in patients without the gBRCA mutation (OR 2.72, 95% CI 1.71–4.32, *p* < 0.001). In addition, an overall higher pCR rate was observed for the platinum-free regimen among gBRCA mutation carriers (54.3%) compared with non-carriers (32.7%) [30]. However, a limited number of BRCA-mutated patients (N = 96) was included in this analysis. Moreover, in the GeparSixto trial, patients received non-standard NACT containing non-standard anthracycline (administered at a low dose) not including an alkylating agent. Therefore, no firm conclusions could be drawn in this regard.

The GeparSixto GBG66 and the BrighTNess studies were also included in a recent meta-analysis of seven phase II-III clinical trials assessed to establish if the presence of a gBRCA mutation could improve the pCR rate in TNBC patients receiving platinum-based NACT. This study showed that the addition of a platinum compound to NACT increases the pCR rate in BRCA-mutated as compared to wild-type TNBC patients, although this difference was not statistically significant (OR = 1.459 CI 95% [0.953–2.34] *p* = 0.082) [32]. However, clinical trials evaluated in this review included different types of platinum-based NACTs in terms of the number of cycles, frequency, and drugs administered (including PARP inhibitors), and the overall number of BRCA-mutated patients was limited (N = 159). Moreover, only two RCTs were included in this meta-analysis. Therefore, despite the OR obtained suggesting a clinical benefit with the addition of platinum compounds in BRCA-mutated TNBC patients, the authors concluded that more data are needed to definitely confirm the utility of platinum compounds in this setting.

In the final analysis, our study confirms the peculiar chemosensitivity of BCs carrying the BRCA-mutation, reporting increased activity of platinum compounds in this patient population. The pCR rate observed in gBRCA mutation carriers receiving a platinum-based NACT is in line with those of other clinical trials. Conversely, the non-alkylating NACT backbone administered in the GeparSixto GBG66 trial does not allow us to make comparisons with this study, especially in patients without gBRCA mutation. In addition, in our study, only 13 patients with a gBRCA mutation received platinum-free NACT. However, the aim of our analysis was not to evaluate the addition of platinum to NACT in TNBC patients but to better define the role of BRCA mutation status within this patient population.

Regarding the analysis of long-term outcomes, our study showed no significant difference in terms of the EFS rate and the OS rate between gBRCA mutation carriers and non-carriers. Patients receiving a platinum-based NACT seem to experience better EFS compared with patients treated with a platinum-free regimen, although this result was not statistically significant. Notably, among BRCA-mutated patients, only two (7.7%) events were reported in patients who received a platinum compound, whereas five events (38.4%) were observed in patients who received platinum-free NACT. Despite the fact that this trend did not achieve statistical significance, these findings may suggest a clinical benefit from the addition of a platinum compound to NACT in patients with the gBRCA mutation.

A secondary analysis of the GeparSixto GBG66 trial reported long-term outcome results of TNBC patients receiving platinum-based NACT according to gBRCA mutational status [39]. After a median follow-up of 35 months, the addition of the carboplatin-improved Disease-Free Survival (DFS) rate in both BRCA mutation carriers and non-carriers (HR 0.55; 95% CI, 0.32–0.95; *p* = 0.03), but the benefit was observed only in the wild-type cohort, in which DFS increased from 73.5% to 85.3% in the carboplatin group (HR 0.53; 95% CI, 0.29–0.96; *p* = 0.04). Conversely, the addition of carboplatin did not improve the DFS rate among gBRCA1/2 mutation carriers (82.5% vs. 86.3% in carboplatin-treated vs. untreated patients) who did, however, experience a better DFS compared to non-carriers. Nevertheless, the different treatment schedules and the different NACT backbones, as mentioned above, do not allow us to compare these data to our findings.

Recently, long-term outcome results of the BrighTNess study were also presented, confirming the clinical benefit of adding carboplatin to NACT in TNBC patients [40].

Finally, in our study, we observed that patients who achieved pCR had significantly improved EFS and OS rates than patients who did not, regardless of BRCA1/2 mutation status and the type of NACT received. Consistently with previous studies, these findings confirm the clinical benefit of achieving pCR, which should always be the target for TNBC patients receiving NACT [13,39,40,41].

Despite the limited number of gBRCA1/2 mutation carriers included in our study, a longer follow-up period could help to better define the possible impact of BRCA mutation status on long-term outcomes in TNBC patients receiving NACT.

Apart from patients with pCR, in the future, it will be important to focus on patients who are refractory to platinum-based treatments and to better understand the mechanisms of action and resistance of these drugs. In fact, literature data show that likely only 1–10% of intracellular cisplatin enters the nucleus and interacts with DNA, causing apoptosis in rapidly proliferating tumor cells. Therefore, various scenarios related to mechanisms of action different from the most well-known ones open up, such as acidification of the cytoplasm and the disruption of RNA transcription. In this regard, a new type of platinum anti-cancer drug is being developed, thanks to the application of nanotechnology, that allows the creation of platinum-nanodrugs (Pt NCs) ions that cause irreversible DNA damage. One problem with treatment with platinum salts is toxicity, which represents a limit of clinical application; in this regard, targeting peptides with the ultimate goal of reducing their toxicity are in development [42]. In fact, the mechanisms of sensitivity and resistance to platinum derivatives, and also to PARP inhibitors, are not yet fully understood, both from genetic and epigenetic points of view. There are numerous studies on the genomic target of these drugs; however, the genomic target may not be the only factor responsible for drug sensitivity: Single gene-drug associations were only rarely able to explain the range of drug sensitivities observed in cell line studies. Few data are available on secondary pharmacological effects such as acidification of the cytoplasm, disruption of RNA transcription, and inhibition of important oncogenic proteins [40]. Therefore, there is difficulty in the interpretation of the “functional” mechanism of action rather than genomic (for example, the modulation of cellular plasticity secondary to treatment) [43]. In this retrospective study, patients with TNBC only underwent BRCA testing according to international guidelines in order to build a therapeutic and prophylactic surgical algorithm in the case of the gBRCA pathogenic variant. Since this is standard management, no pathogenic variants of homologous recombination deficiency (HRD) genes have been evaluated according to ESCAT evidence (tier I—routine use) [44].

In our single-institution real-life study, we retrospectively analyzed 139 consecutive patients who all attended a common clinical pathway with the same surgeons, oncologists, geneticists, and pathologists. Therefore, the patient population was observed homogeneously. However, there are some limitations in our analysis. Despite the population being equally distributed in relation to the NACT regimen received (platinum-based vs. platinum-free), the non-randomized nature of the study may have influenced our results, albeit patients’ characteristics appeared to be homogeneous in these two subgroups. The limited number of BRCA-mutated patients may also have affected the outcome of statistical analysis, and, in addition, our study included patients with stage I TNBC, who were excluded in the main RCTs evaluating NACT efficacy among gBRCA-mutation carriers. While this may not necessarily be considered a limitation of the study, it certainly makes comparisons with other clinical trials more difficult.

Furthermore, in the long-term outcome evaluation, some not reported variables (type of surgery, eventual adjuvant chemotherapy, and/or radiotherapy) may have influenced our findings. As stated before, a longer follow-up would be necessary to draw solid conclusions regarding the survival outcomes of the patients.

## 5. Conclusions

Our analysis shows that TNBC patients with the deleterious gBRCA1/2 mutation have significantly higher pCR rates than patients without the gBRCA1/2 mutation. The benefit of harboring a gBRCA mutation was confirmed only in the subset of patients who received platinum-based NACT, supporting the use of this chemotherapy regimen in this patient population. Moreover, TNBC patients who achieved pCR after NACT had significantly better EFS and OS than patients who did not, regardless of BRCA1/2 mutation status and type of NACT received. Long-term follow-up analyses are needed to further define the impact of gBRCA mutation status on survival outcomes in TNBC patients receiving NACT.

## Figures and Tables

**Figure 1 cancers-14-04571-f001:**
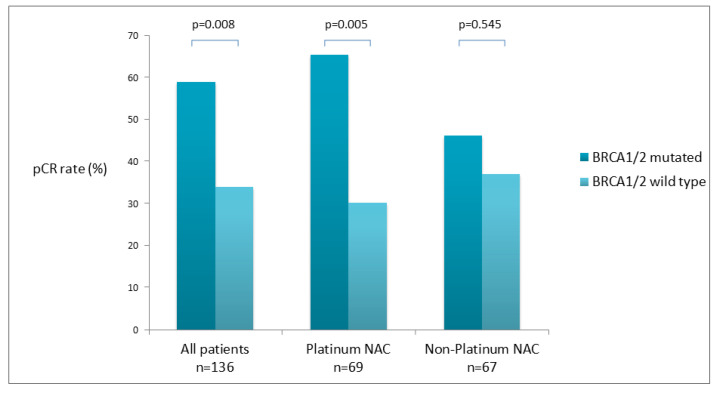
Pathologic complete response (pCR) according to germline BRCA mutation status.

**Figure 2 cancers-14-04571-f002:**
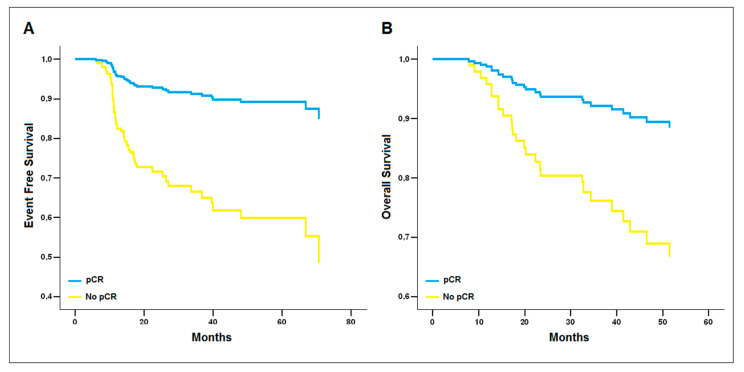
Reduced model of the multivariate Cox regression analysis. pCR is the only covariate associated with improved (**A**) Event-Free Survival (β = 1.50 ± 0.45, *p* = 0.001, OR = 4.5 [1.9–10.7]) and (**B**) Overall Survival (β = 1.20 ± 0.50, *p* = 0.016, OR = 3.3 [1.3–8.9]).

**Table 1 cancers-14-04571-t001:** Demographic and disease characteristics of the patients.

	Overall Patients N° (%)
**Patients (N°)**	136
**Age, years**	
<40 40–59 ≥60 Median (range)	41 (30.2%) 80 (58.8%) 15 (11%) 46 (25–77)
**Menopausal status**	
Postmenopausal Premenopausal	64 (47.1%) 72 (52.9%)
**Histology**	
Ductal Lobular Other	130 (95.6%) 3 (2.2%) 3 (2.2%)
**Histological primary tumor grade**	
High Intermediate Low Unknown or not reported	104 (76.5%) 27 (19.8%) 0 (0.0%) 5 (3.7%)
**Ki-67**	
>20% ≤20% Not reported	132 (97%) 2 (1.5%) 2 (1.5%)
**Clinical TNM Staging (AJCC 8th Ed.)**	
Stage I Stage II Stage III	16 (11.8%) 79 (58.1%) 41 (30.1%)
**Clinical lymph node stage**	
cN0 cN1 cN2 cN3	59 (43.4%) 50 (36.7%) 16 (11.8%) 11 (8.1%)
**Neoadjuvant chemotherapy**	
Carboplatin-containing regimen Non-carboplatin-containing regimen	69 (50.7%) 67 (49.3%)
**Germline BRCA status**	
Deleterious mutation in BRCA1 or BRCA2 No deleterious mutation in BRCA1 or BRCA2	39 (28.7%) 97 (71.3%)

**Table 2 cancers-14-04571-t002:** Demographic and disease characteristics among BRCA1/2 mutation carriers.

**Age at Diagnosis**	
Median (range)	40 (28–65)
**Germline BRCA status**	
Mutation in BRCA1 Mutation in BRCA2 No mutation in BRCA1 or BRCA2	32 (23.6%) 7 (5.1%) 97 (71.3%)
**Histological primary tumor grade**	
High Intermediate Low	32 (82%) 7 (18%) 0 (0%)
**Ki67**	
>20% ≤20% Not reported	36 (92.3%) 1 (2.6%) 2 (5.1%)
**Lymph node stage**	
cN0 cN1 cN2 cN3	16 (41.0%) 14 (35.9%) 6 (15.4%) 3 (7.7%)
**Neoadjuvant chemotherapy**	
Carboplatin-containing regimen Non-carboplatin-containing regimen	26 (66.7%) 13 (33.3%)

**Table 3 cancers-14-04571-t003:** Spectrum of germline BRCA1 and BRCA2 mutations identified. Among BRCA1 mutations observed, deletions of whole exons (17–18; 20–21) were also found (not shown).

BRCA Gene	Exon	HGVS Nucleotide	HGVS Protein	RefSNP	Type	Class
1	5	c.181T>G	p.Cys61Gly	rs28897672	M	5
1	8	c.514delC	p.Gln172AsnfsTer62	rs80357872	F	5
1	11	c.1360_1361delAG	p.Ser454Ter	rs80357969	F	5
1	11	c.5351_5352dupA	p.Asn1784LysfsTer3	rs80359507	F	5
1	11	c.1217dupA	p.Asn406LysfsTer6	rs397508846	F	5
1	11	c.3514G>T	p.Glu1172Ter	rs397509079	NS	5
1	11	c.3285delA	p.Lys1095AsnfsTer14	rs397509051	F	5
1	12	c.4117G>T	p.Glu1373Ter	rs80357259	NS	5
1	13	c.4357insT	Ala1453ValfsX9/Ala1453GlnfsX3	no rs	F	5
1	16	c.4964_4982del	p.Ser1655TyrfsTer16	rs80359876	F	5
1	17	c.5030_5033delCTAA	p.Thr1677IlefsTer2	rs80357580	F	5
1	17	c.5062_5064delGTT	p.Val1688del	rs80358344	IFDEL	5
1	18	c.5123C>A	p.Ala1708Glu	rs28897696	M	5
1	19	c.5209A>T	p.Arg1737Ter	rs80357496	NS	5
1	20	c.5266dupC	p.Gln1756ProfsTer74	rs397507247	F	5
1	20	c.5263_5264insC	p.Ser1755delinsSerProfs	rs1135401936	F	5
1	22	c.5353C>T	p.Gln1785Ter	rs80356969	NS	5
1	23	c.5431C>T	p.Gln1811Ter	rs397509283	NS	5
2	8	c.658_659delGT	p.Val220IlefsTer4	rs80359604	F	5
2	10	c.1796_1800delCTTAT	p.Ser599Ter	rs276174813	NS	5
2	11	c.3331_3334delCAAG	p.Gln1111AsnfsTer5	rs80357701	F	5
2	11	c.4803dupT	p.Lys1602Ter	no rs	NS	4
2	11	c.5946delT	p.Ser1982ArgfsTer22	rs80359550	F	5

HGVS: Human Genome Variation Society nomenclature; RefSNP: Reference Single-nucleotide polymorphism; M: Missense; NS: Nonsense; F: Frameshift; IFDEL: Inframe deletion; LGR: Large genomic rearrangement 5: Pathogenic variant; 4: Likely pathogenic variant; this classification was also attributed to those novel variants with a canonical deleterious effect (terminator or frameshift) [35].

**Table 4 cancers-14-04571-t004:** Univariate analysis of pathologic complete response (pCR).

Variable	Hazard Ratio (95% CI)	*p*-Value
**BRCA status**		
Mutated vs. Wild type	2.78 (1.29–5.98)	0.0085
**Neoadjuvant chemotherapy**		
Platinum vs. Non-platinum	1.21 (CI 0.61–2.4)	0.58
**TNM Stage**		
1 vs. 2 1 vs. 3 2 vs. 3	2.91 (0.92–9.16) 6.00 (1.69–21.21) 2.06 (0.90–4.68)	0.0678 0.0054 0.0847
**Lymph node status**		
Negative vs. Positive	1.61 (0.80–3.20)	0.1749
**Histological grade**		
G2 vs. G3	1.88 (0.75–4.68)	0.1737

## Data Availability

Data are available on request from the authors.

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
