# Peer review of "BRCA Mutation Status in Triple-Negative Breast Cancer Patients Treated with Neoadjuvant Chemotherapy: A Pivotal Role for Treatment Decision-Making"

_cancers, 2022, doi:10.3390/cancers14194571_

Round 1
Reviewer 1 Report
This manuscript by Pavese et al. evaluated the role of germline BRCA1/2 mutation status on outcomes in TNBC patients treated with neoadjuvant chemotherapy. They found that patients with BRCA1/2 mutation had higher pCR rate than non-mutated patients who received a platinum-based NACT. That’s interesting. However, there are a few concerns for the current version. 1. What determines whether a patient was receiving platinum-based therapy, BRCA mutation status, or characteristics of the patients? Carboplatin-containing regimen is more common in BRCA mutation patients(26/39) than in non-mutation patients(43/97). 2. In this study, the 136 TNBC patients was consecutive or selected, the frequency of BRCA deleterious germline mutation can be as high as 28.7%(39/ 136) in TNBC?
Author Response
This manuscript by Pavese et al. evaluated the role of germline BRCA1/2 mutation status on outcomes in TNBC patients treated with neoadjuvant chemotherapy. They found that patients with BRCA1/2 mutation had higher pCR rate than non-mutated patients who received a platinum-based NACT. That’s interesting. However, there are a few concerns for the current version.
- What determines whether a patient was receiving platinum-based therapy, BRCA mutation status, or characteristics of the patients? Carboplatin-containing regimen is more common in BRCA mutation patients(26/39) than in non-mutation patients(43/97).
Dear Reviewer,
thanks for the interesting comments you made. The patients were not selected to receive platinum-based chemotherapy, in fact the enrollment was consecutive and reflects the modification of the prescribing habits of our unit. The first patients in this cohort were those who underwent standard EC chemotherapy for 4 cycles followed by taxanes. Subsequently, in consideration of the results highlighted by the published studies with platinum-based treatment, we began to carry out this treatment in a standard way.
Despite the population was equally distributed in relation to the NACT regimen received (platinum-based vs platinum-free), the non-randomized nature of the study may have influenced our results, albeit patients' characteristics appeared to be homogeneous in these two subgroups. The advantage of this study is to be monocentric with homogeneous medical and surgical treatment. The downside is that it is a retrospective and non-randomized study.
- In this study, the 136 TNBC patients was consecutive or selected, the frequency of BRCA deleterious germline mutation can be as high as 28.7%(39/ 136) in TNBC?
The patients were consecutive. In our study population the percentage of patients harboring a gBRCA1 / 2 mutation was slightly higher than expected (28.6%). However our Department is also a referral center for the treatment of ovarian cancer, therefore a large number of BRCA1 / 2 mutation carriers (affected and non-affected) perform breast screening programs at our institute. All this allows us to carry out a greater number of genetic tests and also to carry out a large number of TNBC diagnoses at an early stage.
Thanks for the deepening.
Reviewer 2 Report
The authors have presented an interesting research on TNBC patients treated with platinum-based neoadjuvant chemotherapy and the impact of BRCA mutation on the response towards the treatment. The sample size of patient data used in their study is well significant. However there are some minor comments for the authors to address.
1. The authors should explain the abbreviation terms in full before using it, such as the term ‘pCR’ included in the abstract. It is first explained in line 98.
2. The authors should include some recent findings of similar novel genes which are responsible and could be used as a marker for therapeutic response in triple negative breast cancer patients similar to their observation of BRCA. Few of such genes are LXRalpha (https://doi.org/10.1038/s41388-021-01720-w); ITGA7 (https://doi.org/10.1038/s41416-021-01484-w) and also miRNA 195 target SEMA6D.
3. Although the authors have studied the effect using platinum based NACT on TNBC patients with BRCA mutations but could the author throw some light on the effect of epirubicin as NACT on similar study?
4. It would be interesting for readers if authors could include in their discussion, if there are any potential scope of improving the NACT response in patients using targeted therapeutics of the same platinum drugs.
Author Response
the authors have presented an interesting research on tnbc patients treated with platinum-based neoadjuvant chemotherapy and the impact of brca mutation on the response towards the treatment. the sample size of patient data used in their study is well significant. however there are some minor comments for the authors to address.
- the authors should explain the abbreviation terms in full before using it, such as the term ‘pcr’ included in the abstract. it is first explained in line 98.
thank you for your suggestion, done
- the authors should include some recent findings of similar novel genes which are responsible and could be used as a marker for therapeutic response in triple negative breast cancer patients similar to their observation of brca. few of such genes are lxralpha (https://doi.org/10.1038/s41388-021-01720-w); itga7 (https://doi.org/10.1038/s41416-021-01484-w) and also mirna 195 target sema6d.
thanks for this suggestion, we will certainly evaluate the negative modulation of semad by microrna-195 and microrna-26b with consequent resistance to chemotherapy in the future. at this time we wanted to carry out a single-center retrospective critical review of patients with early tnbc treated at our center with and without pathogenic variant of brca.
- although the authors have studied the effect using platinum based nact on tnbc patients with brca mutations but could the author throw some light on the effect of epirubicin as nact on similar study?
in our study no significant difference in pcr rate was observed between patients receiving a platinum-based nact and those receiving a anthra-taxanes regimen (43.5% vs 38.8%; or 1.21; 95% ci 0.61 - 2.4; p=0.58). the difference is noted among patients who received a platinum-containing nact, 65.4% of patients with a deleterious germline brca1/2 mutation achieved a pcr versus 30.2% of patients without a germline brca1/2 mutation (or 4.35, 95% ci 1.54 - 12.30; p= 0.0054).
- it would be interesting for readers if authors could include in their discussion, if there are any potential scope of improving the nact response in patients using targeted therapeutics of the same platinum drugs.
thank you for your suggestion, done
Reviewer 3 Report
Major revision
In this report, the authors investigate a series of BRCA1/2 triple negative breast cancers and conclude that the ones harboring BRCA mutations respond better to therapy. This is an interesting study and connects well with the available literature. However, the authors fail to place their study in light of recent findings. The authors should be aware that in BRCA1/2 mutants double strand break repair relies on RAD52 which is generally error prone. Thus, these findings make sense. The authors should either reintroduce their paper in light of available data or discuss these papers in their “discussion” section. Here are some references that must be considered (PMID: 21148102, PMID: 33784323, PMID: 26873923, PMID: 33716297, PMID: 30590106, PMID: 22964643, PMID: 32175645, PMID: 32255263)
Minor revisions
1. Line 44: “…data from patients with triple negative breast cancer patients…” Why use “patients” twice?
2. Line 47: Please define pCR acronym before using it first. PCR can mean something else. Also, gBRCA.
3. Line 102: There is an error.
4. Please inserted your references before the period of each sentence, not outside the period.
5. Line 185. Why are the authors considering the variants of uncertain significance wild type? I would suggest restratifying by these variants as well.
6. Table 3. There is no reason to put the amino acid change in parenthesis. The proper genetic nomenclature is p.Cys61Gly not p.(Cys61Gly). Also, please indicate how the pathogenicity classes were defined. Did the authors use the ClinVar characterization or some other characterization? Generally, a pathogenicity characterization should be given in p-values (e.g., FATHMM, CHASM or VEST scores). More description should be given in the legend on what the “Class” means if the authors use this qualification.
Author Response
Major revision
In this report, the authors investigate a series of BRCA1/2 triple negative breast cancers and conclude that the ones harboring BRCA mutations respond better to therapy. This is an interesting study and connects well with the available literature. However, the authors fail to place their study in light of recent findings. The authors should be aware that in BRCA1/2 mutants double strand break repair relies on RAD52 which is generally error prone. Thus, these findings make sense. The authors should either reintroduce their paper in light of available data or discuss these papers in their “discussion” section. Here are some references that must be considered (PMID: 21148102, PMID: 33784323, PMID: 26873923, PMID: 33716297, PMID: 30590106, PMID: 22964643, PMID: 32175645, PMID: 32255263)
Minor revisions
- Line 44: “…data from patients with triple negative breast cancer patients…” Why use “patients” twice?
THANK YOU FOR YOUR SUGGESTION, DONE
- Line 47: Please define pCR acronym before using it first. PCR can mean something else. Also, gBRCA.
THANK YOU FOR YOUR SUGGESTION, DONE
- Line 102: There is an error.
THANK YOU FOR YOUR SUGGESTION, DONE
- Please inserted your references before the period of each sentence, not outside the period.
THANK YOU FOR YOUR SUGGESTION, DONE
- Line 185. Why are the authors considering the variants of uncertain significance wild type? I would suggest restratifying by these variants as well.
THANK YOU FOR YOUR SUGGESTION, DONE
- Table 3. There is no reason to put the amino acid change in parenthesis. The proper genetic nomenclature is p.Cys61Gly not p.(Cys61Gly). Also, please indicate how the pathogenicity classes were defined. Did the authors use the ClinVar characterization or some other characterization? Generally, a pathogenicity characterization should be given in p-values (e.g., FATHMM, CHASM or VEST scores). More description should be given in the legend on what the “Class” means if the authors use this qualification.
THANK YOU FOR YOUR SUGGESTION, DONE
Round 2
Reviewer 1 Report
I do not have any further comments.
Reviewer 3 Report
The authors have made changes. This reviewer is satisfied.